# Metagenomic chromosome conformation capture (meta3C) unveils the diversity of chromosome organization in microorganisms

Martial Marbouty[1,2], Axel Cournac[1,2†], Jean-François Flot[3†], Hervé Marie-Nelly[1,2†], Julien Mozziconacci[4], Romain Koszul[1,2*]

[1]Groupe Régulation Spatiale des Génomes, Département Génomes et Génétique, Institut Pasteur, Paris, France; [2]Centre national de la recherche scientifique, UMR 3525, Paris, France; [3]Biological Physics and Evolutionary Dynamics Group, Max Planck Institute for Dynamics and Self-Organization, Göttingen, Germany; [4]Department of Physics, Laboratoire de physique théorique de la matière condensée, Université Pierre et Marie Curie, Paris, France

**Abstract** Genomic analyses of microbial populations in their natural environment remain limited by the difficulty to assemble full genomes of individual species. Consequently, the chromosome organization of microorganisms has been investigated in a few model species, but the extent to which the features described can be generalized to other taxa remains unknown. Using controlled mixes of bacterial and yeast species, we developed meta3C, a metagenomic chromosome conformation capture approach that allows characterizing individual genomes and their average organization within a mix of organisms. Not only can meta3C be applied to species already sequenced, but a single meta3C library can be used for assembling, scaffolding and characterizing the tridimensional organization of unknown genomes. By applying meta3C to a semi-complex environmental sample, we confirmed its promising potential. Overall, this first meta3C study highlights the remarkable diversity of microorganisms chromosome organization, while providing an elegant and integrated approach to metagenomic analysis.

*For correspondence: romain.koszul@pasteur.fr

†These authors contributed equally to this work

## Introduction

Microbial species have for a long time been studied individually, leading to the development of applications in fields as diverse as agronomy, environment, or medicine. Sequencing and analyzing the genetic material of microbial communities directly collected from various natural environments such as skin, gut, soil, and water have dramatically improved our knowledge and understanding of their diversity and interrelations (*Handelsman et al., 1998*; *Guermazi et al., 2008*; *The Human Microbiome Jumpstart Reference Strains Consortium, 2010*; *Mackelprang et al., 2011*; *Lundberg et al., 2012*; *Le Chatelier et al., 2013*). A number of techniques have been developed to improve the resolution and accuracy of individual genome assembly in mixed populations, for instance by separating the species prior to sequencing (*Fitzsimons et al., 2013*) or by refining the clustering and scaffolding procedures used to process sequence data (*Albertsen et al., 2013*). However, these approaches remain generally limited by the use of complex technologies and/or by the need to construct multiple libraries. In parallel, an interesting development in the field of genome assembly has recently arisen from the realization that the physical properties of chromosomes contain information regarding their linear structure. The frequency with which two chromosomal segments come in contact, as measured

**eLife digest** Microbial communities play vital roles in the environment and sustain animal and plant life. Marine microbes are part of the ocean's food chain; soil microbes support the turnover of major nutrients and facilitate plant growth; and the microbial communities residing in the human gut support digestion and the immune system, among other roles. These communities are very complex systems, often containing 1000s of different species engaged in co-dependent relationships, and are therefore very difficult to study.

The entire DNA sequence of an organism constitutes its genome, and much of this genetic information is stored in large structures called chromosomes. Examining the genome of a species can provide important clues about its lifestyle and how it evolved. To do this, DNA is extracted from cells and is then usually cut into smaller fragments, amplified, and sequenced. The small stretches of sequence obtained, called reads, are finally assembled, yielding ideally the complete genome of the organism under study.

Metagenomics attempts to interpret the combined genome of all the different species in a microbial community and has been instrumental in deciphering how the different species interact with each other. Metagenomics involves sequencing stretches of the community's DNA and matching these pieces to individual species to ultimately assemble whole genomes. While this may be a relatively straightforward task for communities that contain only a handful of members, the metagenomes derived from complex microbial communities are huge, fragmented, and incomplete. This often makes it very difficult or even nearly impossible to match the inferred DNA stretches to individual species.

A method called chromosome conformation capture (or '3C' for short) can reveal the physical contacts between different regions of a chromosome and between the different chromosomes of a cell. How often each of these chromosomal contacts occurs provides a kind of physical signature to each genome and each individual chromosome within it.

Marbouty et al. took advantage of these interactions to develop a technique that combines metagenomics and chromosome conformation capture—called meta3C—that can analyze the DNA of many different species mixed together. Testing meta3C on artificial mixtures of a few species of yeast or bacteria showed that meta3C can separate the genomes of the different species without any prior knowledge of the composition of the mix. In a single experiment, meta3C can identify individual chromosomes, match each of them to its species of origin, and reveal the three-dimensional structure of each genome in the mix. Further tests showed that meta3C can also interpret more complex communities where the number and types of the species present are not known.

Meta3C holds great promise for understanding how microbial communities work and how the genomes of the species within a community are organized. However, further developments of the technique will be required to investigate communities as diverse as those present in most natural environments.

in genome-wide chromosome conformation capture (3C) experiments (*Dekker et al., 2002*; *Lieberman-Aiden et al., 2009*), obeys to some extent the laws of polymer physics (*Rippe, 2001*; *Wong et al., 2012*). Contact frequencies can be used to identify synteny between DNA segments up to a few 100s kilobases (kb) apart, and two studies recently applied this concept to refining the human genome sequence (*Burton et al., 2013*; *Kaplan and Dekker, 2013*). Such approach therefore shows great promise as a general method for improving genome assemblies. Using chromosomal contacts data in order to generate the precise scaffold of a given genome is however a difficult task: hence, we developed GRAAL, a robust and explicit statistical method to tackle this issue at the highest resolution possible (*Marie-Nelly et al., 2014a*; *Figure 1A*). Since this approach enables the identification of individual chromosomes within the same nuclei, we hypothesized that tridimensional (3D) organization holds enough specific information to distinguish as well DNA segments of chromosomes present within different organisms (*Figure 1A*). In other words, if the interactions between chromosomes contained within a human or a fungal nucleus can be resolved, then it may be possible to resolve the chromosomes of various species mixed together. In the present work, we show that this is indeed the case: a single meta3C experiment, performed on a mix of species, allows the de novo assembly

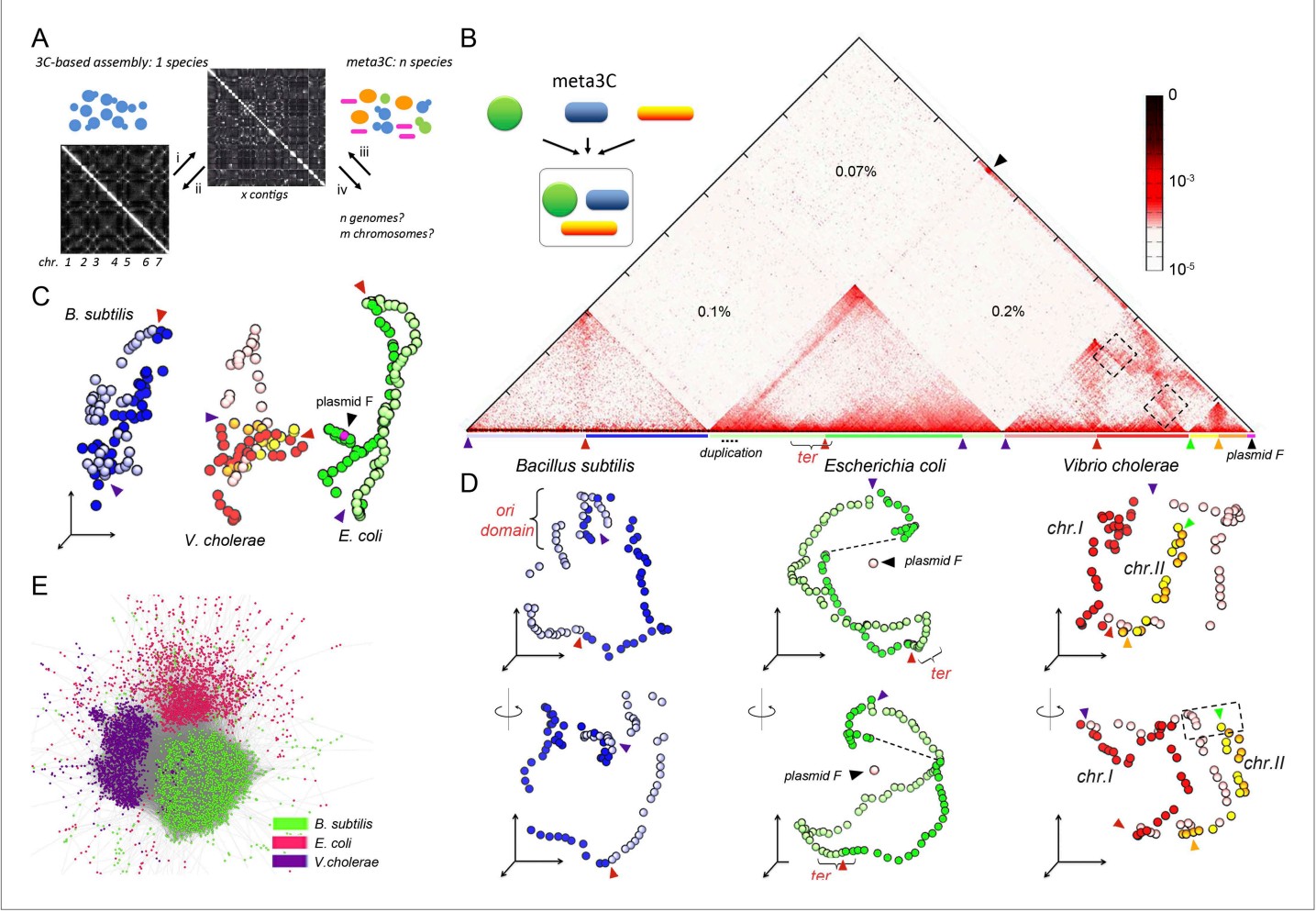

**Figure 1**. meta3C experiment on a controlled mix of bacterial species. (**A**) Schematic representation of the principle of a meta3C experiment. For a single species, one can (i) generate a genome-wide contact map but also (ii) use the genomic contact data to reorder the contact matrix of a poorly assembled genome (top) in order to scaffold its contigs into a more likely structure (bottom left). For a mixture of species, one can similarly generate a contact map directly from the mix (iii) and then use it to characterize the genomes of the species contained in the mixture (iv). (**B**) Chromosomal contact map of a mixture of three bacteria. The darker the shade in the matrix, the higher the contact frequency (color scale in log). Blue and green arrowheads: origins of replication of the chromosomes. Red and orange arrowheads: termination of replication. Each chromosome arm is represented with a color code. The percentages of interactions that occurred between different species are indicated within the matrix. We detected frequent interactions between a F plasmid sequence (black arrowhead) and a large duplicated region of the *E. coli* chromosome (dotted line). (**C**) 3D reconstruction of the entire contact matrix. Each replichore of the four chromosomes is represented with a different color, with the colored arrowheads indicating *Ori* positions. Black arrowhead: F' plasmid and *E. coli* chromosome when contacts with the duplication are taken into account; see *Figure 1—figure supplement 2* for more details. (**D**) Two different views of the 3D reconstruction of each of the three bacteria taken individually. *Ter* and *Ori* are indicated with colored arrowheads (see above). *E. coli* and *B. subtilis* domains are also shown. The F' plasmid is positioned according to its 3D contacts with the *E. coli* genome, without taking into account the duplication (dotted line within the *E. coli* genome). (**E**) Networks of interactions between the different contigs for the mix of three bacteria represented with a force-directed graph-drawing algorithm. Each node represents a contig from the de novo assembly. Each link in gray represents at least one 3C contact. Each color corresponds to one community detected by the Louvain algorithm.

The following figure supplements are available for figure 1:

**Figure supplement 1**. Numbers of intra-specific and inter-specific (chimeric) pairs of reads from the meta3C experiment performed on the bacterial mix.

**Figure supplement 2**. Analysis of the organization of the F' plasmid in the *E. coli* strain used in this study.

**Figure supplement 3**. Pearson correlation matrix of the meta3C bacterial experiment.

and scaffolding of the various genomes present in the mixture without prior knowledge of their genome sequences. We also show that the method allows deciphering the average 3D organization of these genomes in space, unveiling a remarkable diversity of chromosome organization in microorganisms. Therefore, meta3C paves the way to the integrated characterization and analysis of metagenomes in complex populations.

## Results

### Meta3C unveils the diversity of chromosome organization of a mix of three bacterial species

We first processed a controlled mix of three bacterial species—*Escherichia coli* (Gram−), *Vibrio cholerae* (Gram−), and *Bacillus subtilis* (Gram+)—into a metagenomic 3C (meta3C) library and sequenced it on an Illumina platform (Hiseq2000 – Paired End – 2 × 104 bp). The reads were aligned on the three reference genome sequences to generate a chromosome contact map of the whole population (*Figure 1B*). To construct this map, each genome was divided into bins of 30 kb, and the contact frequencies were normalized so that their sum equaled one for each bin (see 'Materials and methods'; *Cournac et al., 2012*). As in a typical 3C experiment squares appeared on the diagonal of the matrix, revealing individual chromosomes: the circular chromosomes of both *E. coli* and *B. subtilis* were recovered as separate entities, whereas the two chromosomes of *V. cholerae* yielded two squares exhibiting higher contact frequencies with each other than with the chromosomes of the other species—as expected since these chromosomes share the same cellular compartment (*Figure 1B*). The background generated by cross-species interactions was low (0.37% of the total interactions), indicating that few chimeric pairs of reads (in which one read came from the genome of one species and the other read from the genome of another species) had been generated during the construction of the library (*Figure 1—figure supplement 1*). The meta3C contact matrix was subsequently converted into a 3D structure ('Materials and methods', *Figure 1C*, *Animation 1*). In this representation, bins are shown as beads and the distance between each pair of beads is optimized in proportion to the inverse of their measured contact frequency (*Lesne et al., 2014*). Using this approach, we recovered three populations of bins that corresponded to the three bacterial genomes, with the two chromosomes of *V. cholerae* being visualized in close vicinity to each other. The continuity and circularity of each genome was clearly apparent in the reconstructed structures.

A sequence corresponding to a F plasmid (the fertility factor of *E. coli*) was characterized in the reads and appeared to interact with the genome of the *E. coli* HB101 strain used in the analysis (*Figure 1B*, black arrow). The 3D data provided us with the opportunity to carefully investigate the structure and spatial positioning of this plasmid with respect to the *E. coli* chromosome. We first noticed that a large (~140 kb) region of the *E. coli* genome exhibited a two fold increase of the read coverage, indicating a segmental duplication. This region was also enriched in contacts with the F plasmid sequence (*Figure 1—figure supplement 2A,B*), prompting us to hypothesize that one of the two copy was actually carried by a F' plasmid (i.e., a F plasmid carrying bacterial sequences). A correlation analysis of the contacts between these two regions of interest (a chromosome region encompassing the duplication and the plasmid) revealed clearly that, on the one hand, a copy of the duplication is in close contact with the plasmid (with an IS2 at one the boundaries [coord. 390,063] and an IS3 within), and that on the other hand the plasmid contacts with the *E. coli* genome drop sharply past the duplication boundaries, in agreement with an integration of one copy of the duplication within a F' plasmid (*Figure 1—figure supplement 2C*). This hypothesis was confirmed experimentally by a Southern blot of a pulsed-field gel (*Figure 1—figure supplement 2D*). Having deciphered the linear structure of the F' plasmid within this strain, we investigated its 3D organization with respect to the *E. coli* genome (*Figure 1—figure supplement 2E*). The frequent contacts each copy of the duplication makes with both the chromosome and the F' plasmid resulted in an artifactual co-localization of the plasmid with the chromosome, since the contacts within each of the copies cannot be discriminated and are all positioned along the genome (see *Figure 1—figure supplement 2E*, i and *Animation 2*). To alleviate this artifact, we generated a contact map were all the contacts involving the duplicated region were removed (*Figure 1—figure supplement 2E*, ii and *Animation 3*). In the corresponding 3D structure, the bin representing the plasmid appeared now well isolated from the chromosome (*Figure 1D*). As a supplementary control, we verified that removing from the genome a region of a size similar to the duplication did not impair the 3D positioning of a DNA segment of a size

similar to the F′ plasmid, positioned within this deleted region (blue dot; *Figure 1—figure supplement 2E*, iii). Interestingly, besides this duplication, two regions along the *E. coli* chromosome appeared enriched in contacts with the F′ plasmid: the replication origin (*Ori*) and, to a lower extent, the termination (*Ter*) regions (*Figure 1—figure supplement 2B*). This indicates that the F′ plasmid is positioned preferentially in the vicinity of these regions in fast growing cells, as expected given the preferred location of the F plasmid in mid, 1/4, and 3/4 positions in the cell, which are also preferred positions for *Ori/Ter* regions during the cell cycle (*Gordon et al., 1997*; *Niki and Hiraga, 1997*). These results illustrate the high specificity of the meta3C approach, which allows the identification of genomic regions belonging to each species, and the power of using DNA physical contacts to decipher complex structures in individual genomes.

The remaining two genomes were also considered individually and their average 3D reconstructions generated, unveiling a remarkable diversity of global chromosomal organization in these three bacterial species during exponential growth (*Figure 1D*, *Animation 4*, and *Animation 5*). Interestingly, this diversity appeared rooted in shared principles. One structural feature shared by these bacterial genomes was the global symmetry of the replichores, that is, the two chromosomal arms between the *Ori* and *Ter*. In *B. subtilis* and *V. cholerae* this symmetry was made clearly visible, in the growth conditions used in this experiment, by the presence of a counter-diagonal in the map (running opposite to the continuous and strong signal that accounts for the contact between adjacent DNA segments; see also the correlation version of the contact map on *Figure 1—figure supplement 3*). This global symmetry as well as this second opposite diagonal, reflecting a longitudinal organization of the two replichores extending along parallel axes within the cell, were also reported for *Caulobacter crescentus* (*Umbarger et al., 2011*; *Le et al., 2013*). The two chromosomes of *V. cholera* were found in two different conformations, with chromosome 1 exhibiting a more 'open' configuration and the replichores of chromosome 2 appearing paired (*Figure 1D*). The two arms of the *E. coli* chromosome exhibited an open 3D structure, with no visible diagonal in the matrix.

Another 3D feature shared by these bacteria consisted in the presence of at least one well-defined domain centered either on the *Ori* or *Ter* region. The 3D reconstruction of the *B. subtilis* genome revealed a relatively compact domain at the *Ori* region that overlapped with the region of the genome containing the centromere-like *parS* sites, in agreement with the structural role described for the ParS/Spo0J/Smc complex in the literature (*Gruber and Errington, 2009*; *Sullivan et al., 2009*). By contrast, the *E. coli* chromosome presented a strong *Ter* domain that was clearly identifiable in the contact map and exhibited fewer contacts with the rest of the genome, reflecting most likely the structuring role of the MatP/*matS* system (*Mercier et al., 2008*). For *V. cholerae*, the *Ter* regions of chromosomes 1 and 2 were in closer proximity than the two *Ori* regions, reflecting the controlled segregation mechanisms of these two unequal-sized chromosomes (*Val et al., 2008*). Interchromosomal contacts did not span the entire length of chromosome 1 but started between the *Ori* macrodomain of chromosome 2 and positions located at about one-third of chromosome 1 arms (black dotted squares, *Figure 1D*). In order to visualize the interplay between replication and genomic organization, we applied a color code to represent the read coverage of the genome (reflecting the average progression of the replication fork and thus the relative timing of replication) atop the genome structure of *V. cholerae* (*Animation 6*). Whereas the two *Ter* regions appeared of the same color, consistent with the fact that both chromosomes I and II achieve replication synchronously (*Rasmussen et al., 2007*), the *Ori* regions presented, as expected, different coverages: chromosome I initiates replication first whereas chromosome II starts only to replicate later. This analysis provides a glimpse on the spatial and temporal articulation of the replication program of *V. cholerae*, which would be an interesting example of organization-dependent function. Overall, the chromosome organizations of the three bacteria analyzed (under the exponential, rich-medium growth conditions used in the experiment) appeared remarkably different but shared similar principles, such as the presence of well-defined domains. In order to see if those shared features are conserved across bacteria species, it will be interesting to use a similar high-resolution 3C approach to investigate the organization of bacterial linear chromosomes.

The low amount of 'background interaction', that is, chimeric religation events between restriction fragments belonging to different species, in the experiment incited us to test whether the genomes of these three bacterial species could be directly assembled de novo from the meta3C data by taking advantage of the presence of ~80% 'regular' paired-end reads in the library. This relatively high percentage results from the fact that, unlike the Hi-C protocol (*Lieberman-Aiden et al., 2009*), meta3C does not involve an enrichment step for religated fragments ('Materials and methods'). Using the

assembly program IDBA-UD (*Peng et al., 2012*), a set of 2,436 contigs was generated from the 3C read pairs (N50 = 55 kb, total length 12.5 Mb). The quality of these contigs was assessed by comparing them with published reference genomes, which showed that the assembled contigs covered respectively 96%, 98%, and 93% of the *V. cholerae*, *E. coli*, and *B. subtilis* reference genomes and that only 0.7% of the contigs (52,373 bp total) were chimeric. The meta3C reads were then realigned on the de novo contigs and the contact information was used to pool the contigs into communities sharing similar contact behavior (hence likely belonging to the same genome) using the Louvain algorithm (*Blondel et al., 2008*). Three communities of contigs were generated, each corresponding to a different species and covering respectively 96, 98, and 92% of the genomes of *V. cholerae*, *E. coli*, and *B. subtilis*, respectively (*Figure 1E*). This shows that meta3C does not only allow convenient high-throughput analysis of the 3D organization of a mix of bacterial species but also provides an efficient way to assemble de novo the genomes of these species.

## Meta3C unveils the conservation of chromosome organization in a mix of 11 yeast species

To see if this approach could be applied to a more complex mix of eukaryotic species, we pooled 11 yeast species and performed a meta3C experiment directly on this mixture (*Figure 2A*). The meta3C contact matrix of the 11 reference genomes put side by side presented discrete squares on the diagonal, each corresponding to a species from the mix (*Figure 2A*; see also *Figure 2—figure supplement 2*). The 3D representation of this contact map revealed, again, a very low level of background interactions in the experiment (*Figure 2B*, *Animation 7*; see also *Figure 2—figure supplement 1*). In each of the squares, the co-localization of centromeres resulting from the Rabl configuration was clearly visible (*Figure 2C*, blue arrowheads; see also *Figure 2—figure supplement 4*; *Duan et al., 2010*; *Marie-Nelly et al., 2014b*). In contrast to the diversity of structures observed for the bacterial species analyzed above, the Rabl configuration appeared as the primary driver of yeast genome organization, as illustrated by the individual 3D structures of the genomes of *Yarrowia lipolytica* and *Naumovozyma castellii* (*Figure 2C*, *Animation 8,9*). To test the potential of using meta3C reads for assembling de novo such a complex mix of yeast genomes, we assembled them using IDBA-UD (N50 = 6,914 bp, total length 138 Mb). The breadth of coverage of the 11 genomes by the resulting contigs ranged from 89.8% (*Candida albicans*) to 98.3% (*N. castellii*), with chimeric contigs (misassemblies) representing ~20% of the total (37 Mb). This high percentage of chimera contrasted with the very low level of misassemblies observed previously for the mix of three bacterial genomes. To monitor the influence of chimeric pairs of reads (originating from intergenomic 3D contacts) on the generation of chimeric contigs, we performed an assembly on the same library using the Velvet software that considers all reads as independents (i.e., without pairing them; *Zerbino and Birney, 2008*). This assembly exhibited a dramatic increase (73%, 69 Mb) in chimeric contigs, demonstrating that these misassemblies were not caused by intergenomic 3C paired-end reads but rather by the frequent occurrence of identical or near-identical genome regions (such as transposable elements) in those eukaryotic genomes. The presence of chimeric contigs in the IDBA_UD assembly did not impede the clustering of the most contigs based on their contact frequencies as determined using the Louvain algorithm. The clustering procedure resulted in 13 sub-groups: one for each of the 11 yeast species, one corresponding to an *E. coli* contaminant, and one comprising various misassembled fragments of mitochondrial genomes. In total, only 1% of the assembly (2% of the contigs) could not be attributed to a given species (*Figure 2D*). A de novo assembly followed by scaffolding using our dedicated program GRAAL (*Marie-Nelly et al., 2014a*) was then applied to the pool of contigs identified as belonging to the genome of *N. castellii*, for which a high-quality reference sequence was available (*Figure 2E*; *Gordon et al., 2011*). Since this algorithm involves a splitting procedure of the contigs into restriction fragments, chimeric contigs were broken into non-chimeric parts thereby correcting the assembly errors mentioned above. After processing, 11 super-scaffolds were recovered, with the reordered contact map presenting the typical co-localized centromeres regions expected from the Rabl configuration (*Figure 2E*; *Animation 10*). Overall these 11 scaffolds covered 94.5% of the reference sequence of the 10 chromosomes (chromosome 3 was split into two scaffolds because of the presence of the large, unassembled rDNA array on this chromosome), illustrating the ability of this de novo analysis to correctly assemble and scaffold unknown genomes (see *Figure 2—figure supplement 3*). The same approach was applied to other species in the mixture, such as *Saccharomyces bayanus* (*Cliften et al., 2003*; *Scannell et al., 2011*;

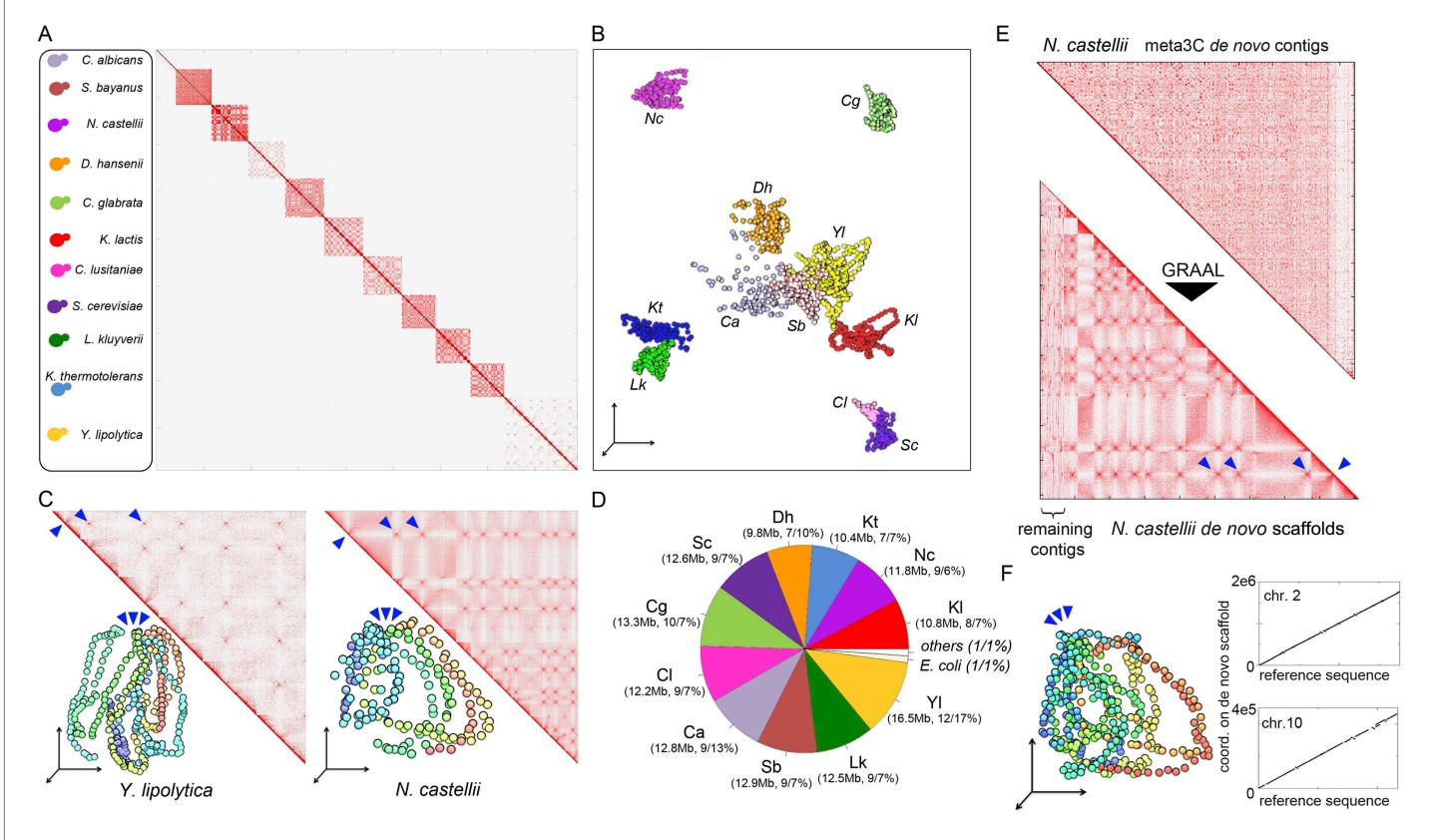

**Figure 2**. meta3C experiment on a controlled mix of yeasts species. (**A**) meta3C contact map of the mix of eleven species. (**B**) 2D projection of the 3D reconstruction of the entire meta3C contact matrix. Each genome occupies an isolated position in space (the 2D projection induces a visual overlap for some species). The color code is the same as for the schematic yeasts on the left panel in (**A**). (**C**) Close-up on the contact maps of three species, with the 3D representation of the matrix in *vis-à-vis*. (**D**) Quantification of the assembly performed using meta3C reads. The first number indicates the amount of total DNA in the community. The two numbers that follow indicates the proportion of the contigs of each community in regards to the total assembly (% of total kb, % of total number of contigs). (**E**) Top: contact map of the contigs present within the community containing mostly sequences from *N. castellii*. The bottom contact map corresponds to the maps recovered following the GRAAL scaffolding. 11 large scaffolds were retrieved, in near-perfect agreement with the known number of chromosomes of this species. Blue triangles: inter-scaffold signal corresponding most likely to the 10 centromeric interactions (*Marie-Nelly et al., 2014b*). (**F**) Corresponding 3D structure of the *N. castellii* de novo meta3C assembly combined with GRAAL processing. The collinearity between two of the resulting superscaffolds and their corresponding reference chromosome sequences is represented in the right panels.

The following figure supplements are available for figure 2:

**Figure supplement 1**. Number of intra-specific and inter-specific (chimeric) pairs of reads in the meta3C contact map.

**Figure supplement 2**. Contact matrices of the genomes of *Y. lipolytica*, *K. thermotolerans*, *N. castellii*, *C. lusitaniae*, *S. bayanus*, *C. glabrata*, *C. albicans*, *K. lactis*, *L. kluyveri*, and *D. hansenii*.

**Figure supplement 3**. Scaffolding using GRAAL of *N. castellii* meta3C contigs.

**Figure supplement 4**. Scaffolding using GRAAL of *S. bayanus* meta3C contigs.

*Figure 2—figure supplement 4*). Our de novo assembly of the genome of *S. bayanus* was more fragmented than the one of *N. castellii*, probably as a result of the lower sequencing depth for this species in the mixture (*Figure 2—figure supplement 4C*; *Animation 11*). Interestingly, not only did de novo assembly and scaffolding from meta3C reads allow genome assembly from a mixed population, but the 3D organization of the assembled genomes was also apparent in the resulting contact maps. For instance, interchromosomal signals corresponding to centromere clustering were clearly

visible in the matrix, emphasizing that 3D signals could also be used to annotate the centromeres of unknown species isolated from meta3C data (*Figure 2F*; *Figure 2—figure supplement 4B*; *Marie-Nelly et al., 2014b*).

## Meta3C allows exploring the genomes of unknown species in a complex environmental sample

As an even more challenging test of the meta3C approach, we confronted it with an environmental sample of unknown and presumably complex composition. For this purpose, sediments were collected from an affluent of the Seine river near Paris. Because environmental genomes usually contain a tremendous diversity of species, including many eukaryotes of large genome sizes, we decided to enrich our sample for prokaryotic organisms by cultivating it for 14 hr in Luria broth and filtrating it prior to meta3C library construction and sequencing. As an internal control of the assembly process, reads from a *V. cholerae* 3C library were added to the meta3C sequences before running IDBA-UD (N50 = 1.2 kb). Contigs generated from the meta3C sediment library were clustered using the Louvain algorithm, yielding 184 significant communities (total size 111 Mb, median size 300 kb, with 19 communities containing more than 1 Mb). The largest 11 ones were included in a contact matrix, revealing again squares along the diagonal (*Figure 3A*). Each community was analyzed using MG-RAST (*Glass et al., 2010*), showing relatively homogeneous taxonomic compositions in most communities (*Figure 3—figure supplement 1*). For each community, more than 80% of the genes identified by MG-RAST within the contigs were attributed to the same taxonomic class. Going down to family level, 8 out of the 11 largest communities were >80% homogeneous, with the remaining three presenting more complex patterns that will require further investigation and development. One community was composed of contigs covering 95% of the *V. cholerae* genome control (for a total amount of 3.9 Mb congruent with the expected size of the genome; *Figure 3B*, i), confirming the ability of this approach to pool DNA regions according to their genome of origin. Other communities contained contigs belonging to other discrete species, related for instance to the ubiquitous bacteria *Aeromonas veronii* or *Exiguobacterium* sp. (for total amounts of 4 Mb and 3.1 Mb, aligning to 72% and 36% of the reference genomes of 4.5 Mb and 3 Mb total sizes, respectively; *Figure 3B*, i). Several species belonging to the classes Bacilli and Enterobacteria were also present in the mix (probably favored by the LB enrichment step). In some instances, weak interactions between communities from these clades suggested that a genome had been split into more than one community or that two communities contained mixtures of closely related species (see dotted square on *Figure 3A*). In spite of this problem the approach seemed to perform relatively well, as reflected by the analysis of the contacts with plasmids. Whether integrated in a genome or under circular forms (see *Figure 1B*), plasmids are expected to present mostly contacts with their host genome since they share the same cellular compartment (see also the contacts of plasmid F with the *E. coli* genome in *Figure 1—figure supplement 1A*). In this study, plasmids annotated as belonging to *Bacillus megaterium* were retrieved almost entirely within a single Bacilli community (*Figure 3B*, ii), suggesting that this community presented indeed a relatively homogeneous content and therefore validating our approach. Although the limited sequencing depth of our experiment restrained our ability to scaffold optimally the contigs of the communities using GRAAL, mapping the reads present in the community related to *A. veronii* against the reference genome of this species revealed a 3D structure reminiscent of those of *B. subtilis* and *C. crescentus* (*Figure 3C*; *Animation 12*). Hence, the 3D information contained within the chromosome of this species was efficiently captured during the meta3C experiment, suggesting that increasing the sequencing depth of a meta3C library will likely make it possible to generate de novo scaffolds and 3D clusters for genomes entirely unknown or underrepresented, shedding light at the same time onto their overall organization.

## Discussion

Our application of meta3C to controlled mixes of bacteria and eukaryotes revealed new 3D genome organizations for four bacterial and several yeast species and showed that this approach can also be used to generate de novo communities of contigs corresponding to individual species in complex metagenomic mixtures (a flowchart of this approach is presented in *Figure 4*).

The average organization of yeast genomes was found to differ radically from those of bacterial nucleoids, most likely because of contrasted replication and division processes and timings. Most of the organization in Saccharomycetes appears driven by the clustering of the centromeres at the

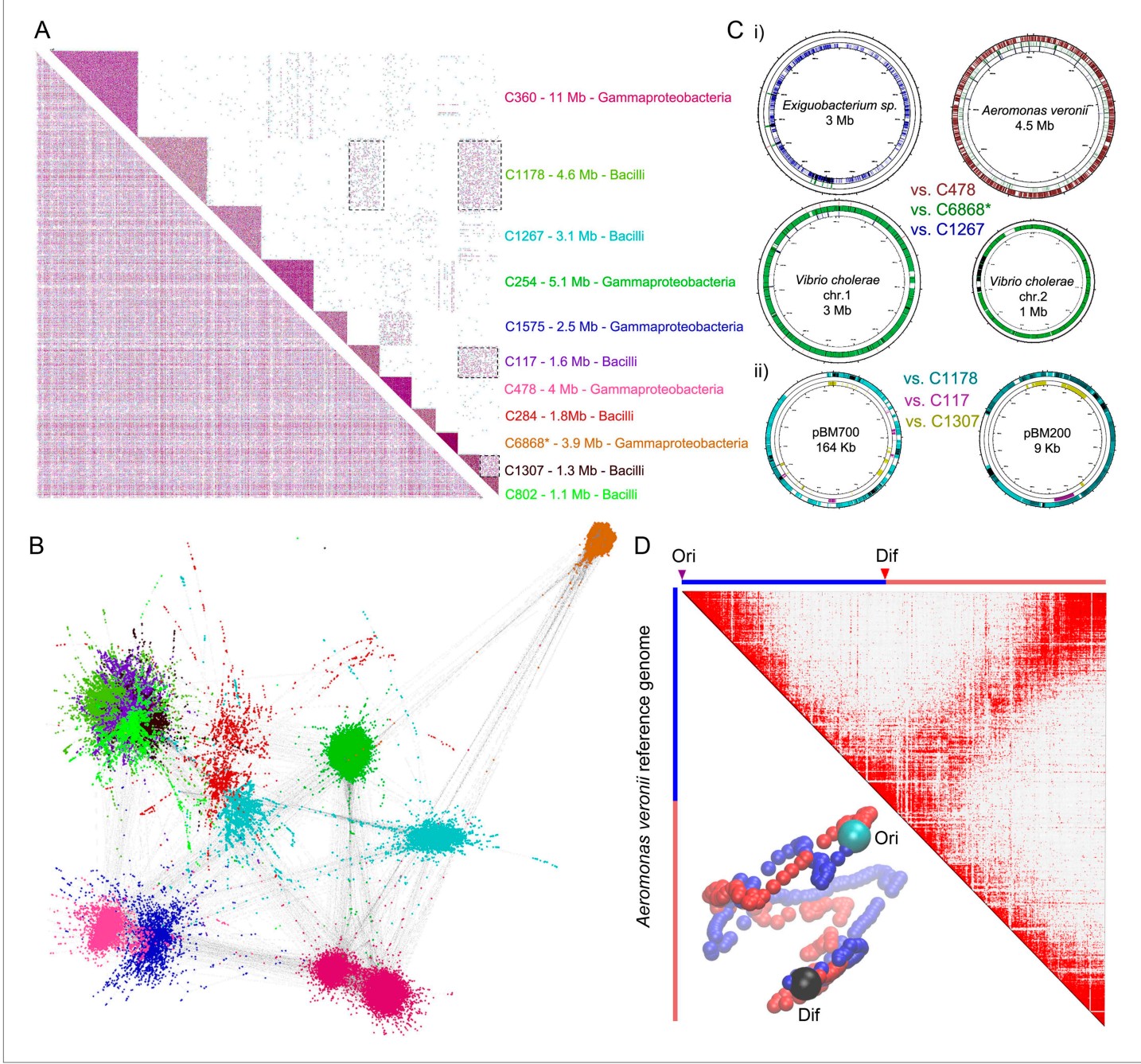

**Figure 3**. meta3C analysis of a complex environmental sample. (**A**) meta3C contact map of the largest 11 communities of contigs in the matrix before (bottom left) and after (upper right) clustering. Each square corresponds to a community grouping contigs that exhibit significantly more contact with each other than with other communities. For each community, an index, the sum of the sequences, and candidate classes are indicated on the right side of the matrix. Dotted square underlines inter-community contact enrichments. C6868*: *V. cholerae*. (**B**) Illustration of the interactions between the 11 largest communities of contigs using a force-directed graph drawing algorithm Force Atlas 2 (*Jacomy et al., 2014*). Each node corresponds to a contig (or a chunk of a contig) and each link represents at least one meta3C interaction. The colors correspond to the communities identified by the Louvain algorithm and described in **A**. (**C**) i: For three communities, the contigs were mapped against the closest reference genomes identified (Exiguobacterium sp. AT1b, *A. veronii* B565, *V. cholerae* N16961, plasmids pBM700 and pBM200). To illustrate the specificity of the approach, each community was mapped against each of those genomes (the color code is the same as in **A** and the order from the outer circle to the inner circle is indicated in the middle). ii: Similarly, the three Bacilli communities highlighted in color were mapped against two plasmid sequences. (**D**) Genomic contact map of an unknown species (most likely *A. veronii* or a close relative) generated by mapping the reads present in the community 478 against the genome of *A. veronii* (bin size = 10 kb). The corresponding 3D structure is indicated next to the contact map.

*Figure 3. Continued on next page*

*Figure 3. Continued*

The following figure supplement is available for figure 3:

**Figure supplement 1**. MG-RAST analysis of the 11 largest meta3C communities.

spindle pole body; this clustering remains the prominent structural feature throughout most of the cell cycle (i.e., G1 + S phase + G2) during exponential growth, with mitosis representing only a fraction of the cycle. In bacteria, in contrary, the strong overall organizational features are the activation of a unique replication origin per chromosome, the ability to initiate multiple replication forks, and the fast division cycle. In addition, important topological constraints are exerted on bacterial circular chromosomes around the *Ter* regions, and our observation of significant contacts at these positions was probably linked to these phenomena. More imaging and 3C-like analyses will be required to understand the precise choreography of chromosome segregation in these different microorganisms.

Taking advantage of chromatin conformation capture data to address genomic questions is a dynamic field: while this paper was under review, two studies were released that also aimed at exploiting the physical contacts between DNA molecules to deconvolve genomes from controlled mixes of microorganisms (***Beitel et al., 2014***; ***Burton et al., 2014***). Although these analyses open interesting

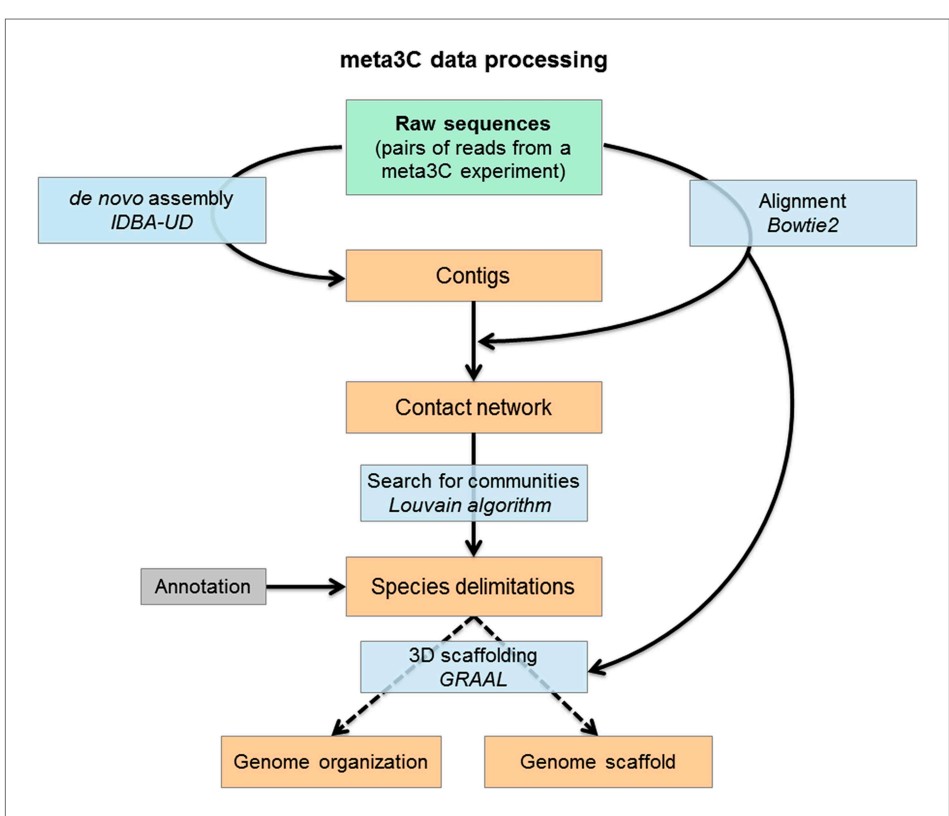

**Figure 4**. Flowchart representing the computational analysis steps of a meta3C experiment. First, the reads from the sequenced meta3C library are assembled into contigs. The meta3C contact information between the contigs is then used to generate a network of the contacts of all contigs against each other. This contact network is searched for so-called 'communities' (by analogy with social network analysis) using the Louvain algorithm. The significant communities can be annotated by NCBI and correspond principally to sequences from individual species. Ultimately, the contact information can be used to reorder the DNA segments within each community and thus generate the 3D contact map of the corresponding species. In this process contigs can be reassembled by software such as GRAAL, thereby decreasing the percentage of chimeric fragments in the assembly and improving its continuity. DNA segments originating from other species are put aside automatically during this process given their lack of 3C contacts with the rest of the genome under reassembly.

perspectives, the meta3C approach presented in this work differs in several aspects. First, both studies used HiC to scaffold genomes from contigs obtained either from simulated assemblies (*Beitel et al., 2014*) or from independent experiments (*Burton et al., 2014*). Our approach conveniently uses a single meta3C library and a blind analysis for generating contigs, binning them, scaffolding them, and revealing the 3D structure of the corresponding species. Such approach based on a single experiment may require more sequencing depth than an approach combining multiple libraries (HiC + shotgun + mate pair, for instance), but the exact trade-off remains to be determined and depends on the specifics of the experiment, the sequencing technology, and on the amount of starting material available.

Besides, our blind analysis allowed us to delineate communities without prior knowledge of the number of species present in the mix; this contrasts with the approach of *Burton et al. (2014)*, which apparently requires such prior knowledge. Similarly, we generate de novo chromosome scaffolds without any assumptions regarding the number of chromosomes present in the mixture, a realistic approach when it comes to exploring environmental samples or complete the assembly of complex genomes.

Furthermore, our experience with the analysis of chromosome organization in both bacteria and yeast species emphasized the importance of the initial steps for the success of such experiment. The adequacy between the cross-linking step, which depends on the incubation time and the concentration of the fixating agent, and the restriction step and choice of the restriction enzyme, is essential for the recovery of long-distance *cis* contacts that improve de novo genome scaffolding and, importantly, reveal the 3D structure. Optimizing the cross-linking conditions allows new insights into the diversity of 3D chromosome organization of several species.

Last but not least, the meta3C approach remains to be applied on a truly natural (not enriched) sample, such as a gut or wine microbiome. Based on our experience, we envision that, this experiment should include several (two at least) enzymes recognizing sites with various GC percentages, in order for both GC-rich and GC-poor genomes to be appropriately represented in the meta3C library. Doing the experiment with multiple enzymes would also allow taking into account GC content (in addition to average coverage) to improve the binning of contigs into communities. Identifying contigs presenting largely divergent characteristics compared to their neighbors and redistributing them to their most likely communities would also lead to improved assemblies for each species (see for instance *Albertsen et al., 2013*). Overall, quantifying physical contacts between chromosomes provides an objective and convenient principle to segregate the genomes of sympatric species and, from there, to explore the biological diversity of complex ecosystems.

## Materials and methods

### Construction of a bacteria meta3C library

Meta3C protocols were adapted from 3C protocols (notably, *Dekker et al., 2002*; *Oza et al., 2009*). The strains used for the meta3C bacterial library were *B. subtilis* BS168 (*Burkholder and Giles, 1947*), *E. coli* HB101 (*Boyer and Roulland-Dussoix, 1969*), and *V. cholerae* MV127 (*Val et al., 2012*). For each strain, 100 ml of LB were inoculated with $10^6$ cells/ml and incubated at 37°C until a final concentration of about $2 \times 10^7$ cells/ml. Cells from the different species were then mixed and cross-linked with fresh formaldehyde for 30 min (3% final concentration; Sigma Aldrich, Saint Louis, Missouri) at room temperature (RT) followed by 30 min at 4°C. Formaldehyde was quenched with a final concentration of 0.25 M glycine for 5 min at RT followed by 15 min at 4°C. Fixed cells were collected by centrifugation, frozen on dry ice, and stored at −80°C until use. Frozen pellets of $3 \times 10^9$ cells were thawed on ice and resuspended in a final volume of 650 µl 1× TE pH 8 before adding 4 µl of Ready-Lyse lysozyme (35 U/µl; Tebu Bio, France), followed by incubation at RT for 20 min. SDS was added to a final concentration of 0.5% followed by 10 min RT incubation. 50 µl of lysed cells were put in eight tubes containing 450 µl of digestion mix (1× NEBuffer 1 [New England Biolabs, Ipswich, Massachusetts], 1% Triton X-100, and 100U *Hpa*II enzyme [NEB; C^CGG]). The chromatin was then digested for 3 hr at 37°C, split into four aliquots, and diluted with 8 ml ligation buffer (1× ligation buffer NEB without ATP, 1 mM ATP, 0.1 mg/ml BSA, 125 units of T4 DNA ligase [5 U/µl − Weiss Units − Thermo Fisher Scientific, Waltham, Massachusetts]). Ligation was performed at 16°C for 4 hr followed by a de-cross-linking step consisting of an overnight (ON) incubation at 65°C in the presence of 250 µg/ml proteinase K in 6.2 mM EDTA. DNA was then precipitated with 800 µl of 3 M sodium-acetate (pH 5.2) and 8 ml iso-propanol. After 1 hr at −80°C, DNA was pelleted by centrifugation. Pellets were

suspended in 500 µl 1× TE buffer and the RNA degraded with a final concentration of 0.03 mg/ml RNAse for 1 hr at 37°C. DNA was transferred into 2 ml centrifuge tubes, extracted twice with 500 µl phenol–chloroform pH 8.0, precipitated, washed with 1 ml cold ethanol (70%), and diluted in 30 µl 1× TE buffer. All tubes were pooled and the resulting 3C library was quantified on gel using the program QuantityOne (Bio-Rad, Richmond, California).

## Construction of a yeast meta3C library

The strains used for the meta3C yeast library were *Y. lipolytica* CLIB122, *L. kluyveri* CBS3082, *Candida lusitaniae* ATCC42720, *C. albicans* SC5T314, *Kluyveromyces lactis* CLIB210, *S. bayanus* 623-6C, *Kluyveromyces thermotolerans* CBS6340, *Saccharomyces cerevisiae* BY4741, *N. castellii* CBS 4309, *Candida glabrata* CBS138, and *Debaryomyces hansenii* CBS767. All strains were grown at 30°C in 50 ml BMW medium until reaching $1 \times 10^7$ cells/ml (*Thompson et al., 2013*). The cultures were then mixed and cross-linked for 30 min with fresh formaldehyde (3%). The formaldehyde was quenched with a final concentration of 0.25 M glycine for 5 min at RT followed by 15 min at 4°C. Fixed cells were pooled as aliquots of $3 \times 10^9$ cells, collected by centrifugation, frozen on dry ice, and stored at −80°C.

Aliquots were thawed on ice and resuspended in 6 ml of 1× *Dpn*II buffer (NEB). The cells were then split into four tubes and lysed using a Precellys grinder (3 cycles: 6700 rpm − 3 × 20 s ON/60 s OFF; Bertin Technologies, France) and VK05 beads. Lysed cells were pooled and their volume was adjusted to 6 ml with 1× *Dpn*II buffer. SDS was added to a final concentration of 0.3% and the solution was split into twelve 2 ml tubes (Eppendorf–DNA LoBind, Eppendorf, Germany) and incubated for 20 min at 65°C followed by 30 min at 37°C under agitation. As a next step, 6 µL of 10× restriction enzyme buffer and 50 µL of Triton X-100 20% were added to each tube, mixed carefully, and incubated at 37°C for another 30 min under agitation. The chromatin was digested for 3 hr with 50 units of restriction enzyme under agitation (*Dpn*II: G^ATC, NEB). Following incubation, 100 units of restriction enzyme were added and the incubation was extended overnight. The digested chromatin was pooled into four equal reactions and the samples were then processed as described above.

## Construction of an environmental meta3C library

The river sediments (300 g) were incubated for one night in 500 ml of LB at 30°C. The next morning, the culture was filtrated on Whatman paper (with a size cut-off of 15 µm) and 20 mg of wet material were resuspended in 100 ml of fresh LB, then treated with fresh formaldehyde (5% final concentration) for 30 min at RT followed by 30 min at 4°C. The formaldehyde was quenched with glycine (0.4 M final) for 5 min at RT followed by 15 min at 4°C. Fixed cells were collected by centrifugation, frozen on dry ice, and conserved at −80°C until use. Frozen pellets were slowly thawed on ice and resuspended in 800 µl of TE 1×. Cells were then lysed using a Precellys grinder (3 cycles: 6700 rpm − 3 × 20 s ON/60 s OFF) and VK05 beads. About 600 µl of lysed cells were recovered, to which SDS was added at a 0.5% final concentration before incubation at RT for 10 min. 50 µl of lysed cells were put in eight tubes containing 450 µl of digestion mix (50 µl tampon NEB 2 10×, 50 µl Triton X-100 10%, 10 µl of *Hae*III enzyme [GG^CC] − 10 U/µl NEB, 340 µl H2O). Chromatin was then digested during 3 hr at 37°C under agitation. The digested chromatin was pooled into four equal reactions and the samples were processed as described above.

## Illumina sequencing

Aliquots of 5 µg of each 3C library were dissolved in water (final volume 130 µl) and sheared using a Covaris S220 instrument (duty cycle 5, intensity 5, 200 cycles per burst, 4 cycles of 60 s each; Covaris Ltd., Woburn, Massachusetts). The sheared DNA was purified on QIAquick columns and processed using a commercial kit (Paired-End DNA sample Prep Kit—Illumina—PE-930-1001; Illumina, San Diego, California). The DNA was ligated to custom-made versions of the Illumina PE adapters (see *Table 1*) for 3 hr at room temperature in a final volume of 30 µl (20 µl of DNA [around 8 µg], 3 µl of ligation buffer 10× [NEB], 3 µl of T4 DNA ligase [400 U/µl from NEB], and 4 µl of 10 µM adapter solutions). Tubes were then incubated at 65°C for 20 min.

DNA fragments ranging in size from 400–800 pb were purified using a PippinPrep apparatus (SAGE Science, Beverly, Massachusetts). For each library, test PCR reactions were performed to determine the optimal number of PCR cycles and a large-scale PCR (eight reactions) was then set-up with the number of PCR cycles determined previously. The PCR products were finally purified using Qiagen MinElute columns (Qiagen, Netherlands) and paired-end (PE) sequenced on an Illumina platform (HiSeq2000; PE 2 × 100).

**Table 1.** List of the custom-made Illumina adapters used in this study

| Oligos | Sequence | Library |
|--------|----------|---------|
| MM70 | GTANNNNNNAGATCGGAAGAGCGGGTTCAGCAGGAATGCCGAG | Mix of 11 yeasts |
| MM71 | ACACTCTTTCCCTACACGACGCTCTTCCGATCTNNNNNNTACT | |
| MM76 | CAGNNNNNNAGATCGGAAGAGCGGGTTCAGCAGGAATGCCGAG | Mix of 3 bacteria |
| MM77 | ACACTCTTTCCCTACACGACGCTCTTCCGATCTNNNNNNCTGT | |
| MM70 | GTANNNNNNAGATCGGAAGAGCGGGTTCAGCAGGAATGCCGAG | River sediment |
| MM71 | ACACTCTTTCCCTACACGACGCTCTTCCGATCTNNNNNNTACT | |

## Processing of PE reads

The raw data from each 3C experiment were processed as follow: first, reads were demultiplexed using the small tag present at the beginning of each sequence (contained in the custom-made adapters). Then, PCR duplicates were collapsed using the six Ns present on each adapter (*Table 1*). Reads from the raw data used in the present study were aligned using Bowtie 2 in its most sensitive mode (*Langmead and Salzberg, 2012*). We used an iterative alignment procedure similar to *Imakaev et al. (2012)*, that is, only the first 20 base pairs of the read were initially mapped then the length of the read was progressively increased until the mapping became unambiguous (with a mapping quality superior to 40). Paired reads were aligned independently. Indexes were built in one step and included the genome sequences of all the different organisms.

## Generation of contact maps

Each mapped read was assigned to a restriction fragment. The matrices were then binned into units of 10 or 200 fragments, resulting in 4292 × 4292 or 2229 × 2229 matrices for the mixtures of three bacteria and 11 yeasts, respectively. Matrices were normalized using the sequential component normalization procedure (SCN) described in *Cournac et al. (2012)*, similar to the iterative normalization procedure described in *Imakaev et al. (2012)*. The SCN procedure ensures that the sum over the column and lines of the matrix equals 1, which reduces the biases inherent to the protocol. Full resolution contact maps are available for bacteria, yeasts, and the environmental sample on the Dryad Digital Repository: http://dx.doi.org/10.5061/dryad.gv595 (*Marbouty et al., 2014*).

## 3D reconstruction of the contact maps

In order to build the 3D structures of the different genomes from the chromosomal contact maps we used the algorithm described in *Lesne et al. (2014)*. Briefly, we first converted the normalized contact matrix into an adjacency graph in which each node represented a genomic region and each link had a weight corresponding to the inverse of the number of contacts detected between the two corresponding nodes in the meta3C experiment. We then converted this graph into a distance matrix using the Floyd–Warshall algorithm. This algorithm computes the distance between each pair of genomic regions by determining the shortest distance on the graph between the two corresponding nodes. We finally converted this distance matrix into a 3D structure using distance geometry theorems as described for molecular structures in *Havel et al. (1983)*. All this procedure was implemented in Matlab.

## Genome assembly

Reads containing undetermined bases were removed before the assembly step to retain only good-quality reads. De novo assemblies were then performed using the program IDBA-UB (*Peng et al., 2012*) with the pre-correction option and default parameters.

## Quantitative analysis of the meta3C library assemblies

### Bacterial meta3C assembly

4,040,422 reads were used for assembling the mixture of three bacterial genomes, yielding 2,436 contigs (303 > 5 kb, N50 = 55 kb, total length 12.5 Mb). More than 95% (~3,880,000) of the initial

reads mapped on the contigs. These contigs were mapped against the reference genomes of the different bacteria using BLAT (**Kent, 2002**). Analyzing the output revealed that 30 and 14 of these contigs covered 96.0% (2,957,563 bp) and 96.1% (1,071,728 bp) of chromosomes 1 and 2 of *V. cholerae*, respectively. Another group of 55 contigs covered 98.4% (4,630,934 bp) of the genome of *E. coli*, whereas a group of 1441 contigs covered 92.5% (3,957,375 bp) of the genome of *B. subtilis* (a higher fragmentation reflecting the lower representation of this bacterium in the mix). We also quantified the amount of chimeric contigs in the assembly using BEDtools (**Quinlan and Hall, 2010**). Such chimeric contigs may occur because of the presence of repeated conserved sequences between bacteria, such as transposons, or because of the background interactions introduced by chimeric religation between restriction fragments belonging to different species. Only 17 out of the 2436 contigs were chimeric (0.7%), accounting for 52,373 bp in total (0.4% of the total size of the assembly). This relatively low number probably results from the limited number of species contained in the mix and their low repeat content. It also shows that chimeric religation during the experiment introduces little background interactions in the assembly process.

## Yeast meta3C assembly

47,614 contigs were generated by assembling the ~60 M yeast reads (7,212 > 5 kb, N50 = 6.9 kb, total length 138.7 Mb). When mapped against the newly generated contigs, 47 M reads (~78%) did align. These contigs were then compared with the reference genomes of the different yeasts using BLAT and the analysis of the output was performed as above. Genome breadths of coverage ranged from 89% (*Y. lipolytica*) to 98% (*N. castellii*). In contrast to the previous results on the bacterial meta3C assembly, there were 9,599 chimeric contigs in the yeast case, representing 27.1% of the total number of contigs. This shows that chimeric contigs arise when dealing with mixtures of repeat-rich genomes, likely because near-identical repeats (such as transposable elements) are found in different genomes in the mixture. Such assembly mistakes will be probably avoided if using longer reads (since repeats longer than the read length pose problem during assembly) or, alternatively, can be resolved a posteriori using GRAAL. Despite the high percentage of chimeric contigs, we were able to assemble ~95% of the *N. castellii* genome from the meta3C reads.

## River sediment meta3C assembly

67,920,671 pairs of reads (91 bp useful) were used to assemble the raw culture from the river sediment. As a positive assembly control, 500,000 reads from a 3C library of *V. cholerae* were added into this pool of sequences. The assembly generated 130,713 contigs (2,250 > 5 kb, N50 = 1,274 kb, total length = 111 Mb). After realignment of the initial reads along the generated contig, 42,705,64 PE reads could be mapped against the assembled contigs (~62%).

## Pulsed-field gel electrophoresis (PFGE)

Chromosome plugs were prepared as described previously (**Koszul et al., 2004**), using lysozyme instead of zymolyase (1% agarose gels, 0.25× TBE buffer at pH8.3). PFGE was performed in a Rotaphor R23 tank (Biometra, Germany) using the following program: 12°C, 5 V/cm for 65 hr, angle 110°, pulse ramps 200 to 80 V. Southern blot hybridization was performed using probes derived from PCR products obtained with the primers listed in *Table 2*.

**Table 2.** List of the oligonucleotides used to PCR amplify probes for the Southern blot of the PFGE (*Figure 1—figure supplement 2*)

| Oligos | Sequence | Coordinates | Probe |
|--------|----------|-------------|-------|
| Meta1 | CATCCCGTGAGAAATAATGGTCG | 2,256,273 (MG1655) | Genomic probe |
| Meta2 | TGTGCATCCCGTCACAAATTC | 2,257,545 (MG1655) | |
| Meta3 | TTGAGCTTATCAAAGTCGTCGGAG | 323,607 (MG1655) | Duplication probe |
| Meta4 | TGATGTGAACTAACGCAGGAAC | 324,796 (MG1655) | |
| Meta5 | TTTACCTCTGATACTGGCTCTGG | 79,024 (plasmid F) | Plasmid F probe |
| Meta6 | ACGTGGCATATTCATGCAGAC | 80,159 (plasmid F) | |

## Binning of contigs using 3C contact data

To group the different contigs into communities reflecting the different genomes present in the sequenced mixtures, we adopted an approach based on graph theory. Among several community-detection algorithms, we found that the Louvain method (*Blondel et al., 2008*) generated the best reconstructions of the controlled mixes of bacteria and yeast species. Before applying the algorithm, contigs longer than 2.5 kb were divided into equal-sized chunks corresponding to several nodes in the graph (since large contigs exhibit more contact than smaller ones, they are more prone to clustering when running the Louvain algorithm). We used the Louvain method with default parameters, except for the mix of three bacteria (r=10 for three communities). For bacterial mix, 10% of the contigs of the smallest size were not attributed to a community with these parameters (representing ~15% of the total reads). For yeast, few contigs were left aside, representing in total ~2% of the number of reads. For the river sediment experiment, ~15% of the contigs were left aside at the binning step. The contigs in each bin resulting from the 3D binning were characterized using BLAST to determine the dominant species.

## GRAAL scaffolding of the *N. castellii* and *S. bayanus* genomes

The program GRAAL (for Genome (Re)-Assembly Assessing Likelihood from 3D) aims at improving incomplete genome assemblies through the probabilistic exploitation of the physical contacts endured by chromosomes within a cellular compartment (*Marie-Nelly et al., 2014a*). The program only needs two datasets for initialization: a set of contigs and a 3C/HiC dataset. Upon initialization and depending on the depth of the HiC dataset, the software splits the contigs into smaller pieces/bins encompassing at least 2 restriction fragments. It then iteratively (over 1000s of steps) searches through a broad range of structures generated from reordering these small bins for genome structures more likely to be true given the 3D data. Importantly, at each step, a bin is tested for a variety of 'structural variations' with respect to its most likely neighbors, including duplications, deletions, fusions, inversion, etc. The likelihood of each of these new genome structures is then computed in light of the contact data, and one of those structures is sampled for the next iteration (during which the position of a new bin is tested). Each bin is tested several times throughout the entire process, and step by step the genome structure originally made of thousands of small independent bins converges towards a structure reflecting the best solution given the 3D contact data.

The entire program and full source code are freely available online here: https://github.com/koszullab/GRAAL as well as on the website of RK laboratory. The description of the program and its application are presented in *Marie-Nelly et al. (2014a)*.

We used GRAAL to reassemble the 2,060 contigs (N50 = 16,764 bp) contained in the *S. bayanus* community identified by the Louvain algorithm. The resulting scaffolding showed a large improvement for all assembly parameters. The N50 length now reached 190,514 bp, and 77% of the total length of the assembly was assembled into regions larger than 50 kb (as compared to 3% before running the software). The result was even more spectacular for *N. castellii* (probably due to its higher coverage in the meta3C library): after running GRAAL on the 999 contigs (N50 = 27,452 bp) in the community delineated by the Louvain algorithm, N50 length reached 792,652 bp, and 96% of the assembled data was in scaffolds larger than 50 kb (with 11 scaffolds covering 95% of the 10 chromosomes).

## Data visualization

For general visualization of the meta3C assembly data we used force-directed graph-drawing algorithms. This class of algorithms positions the nodes of a graph by assigning forces among the set of edges according to weighted interactions between the nodes. In our case, the 3C contacts dictated the strength of interactions between nodes. These layouts allowed us to visualize conveniently large clusters of nodes that were subsequently confirmed by the use of the Louvain algorithm. All graphs were visualized using the network software Gephi (*Bastian et al., 2009*; *Martin et al., 2011*). Contigs longer than 2.5 kbp were divided into equal-sized chunks corresponding to several nodes in the graph (to 'normalize' the appearance of the clusters, since otherwise every contig, whatever its size, would be represented by a single point). For visualizing the meta3C bacterial assembly we used the Force Atlas 2 algorithm (*Jacomy et al., 2014*). This algorithm assigns spring-like attractive forces and repulsive forces (like those between electrically charged particles) to the set of edges and the set of nodes: in the equilibrium state, discrete clusters are obtained. The entire set of bins was then color-labeled,

with each color corresponding to a community detected by the Louvain algorithm (revealing a strong correlation with the clusters obtained using the visualization approach).

For the circular representation and comparison of the sequences present in the communities against known bacterial genomes, we used the CGView server (*Grant and Stothard, 2008*).

## Acknowledgements

We thank Jean-Yves Coppée, Caroline Proux, and Laurence Ma from the Génopole at Institut Pasteur for technical assistance with Illumina sequencing. Thanks also to Nienke Buddelmeijer and Didier Mazel for providing us the *B. subtilis* and *V. cholerae* strains and to Frédéric Boccard, Olivier Espeli, Didier Mazel, Marcelo Nollmann, and Marie-Eve Val for their constructive comments on the results. We are grateful to Agnès Thierry for providing us the yeast strains and for helping us with pulse-field electrophoresis and hybridization of the plasmid F' and the *E. coli* genome. We are also grateful to Olivier Jaillon and to our colleagues from the group Régulation spatiale des génomes for useful comments and discussions. Parts of the computational analyses were performed on the computing cluster of the Gesellschaft für Wissenschaftliche Datenverarbeitung mbH Göttingen (GWDG).

## Additional information

### Competing interests

MM: GRAAL program is owned by the Institut Pasteur, its use for commercial purposes requires a specific licence. AC: GRAAL program is owned by the Institut Pasteur, its use for commercial purposes requires a specific licence. HM-N: GRAAL program is owned by the Institut Pasteur, its use for commercial purposes requires a specific licence. RK: GRAAL program is owned by the Institut Pasteur, its use for commercial purposes requires a specific licence. The other authors declare that no competing interests exist.

### Funding

| Funder | Grant reference number | Author |
|---|---|---|
| European Research Council | 260822 | Romain Koszul |
| Fondation ARC pour la Recherche sur le | 20100600373 | Martial Marbouty |

The funders had no role in study design, data collection and interpretation, or the decision to submit the work for publication.

### Author contributions

MM, AC, Conception and design, Acquisition of data, Analysis and interpretation of data; J-FF, JM, Analysis and interpretation of data, Drafting or revising the article; HM-N, Analysis and interpretation of data; RK, Proposed and coordinated the project, Conception and design, Analysis and interpretation of data, Drafting or revising the article

## Additional files

### Major datasets

The following dataset was generated:

| Author(s) | Year | Dataset title | Dataset ID and/or URL | Database, license, and accessibility information |
|---|---|---|---|---|
| Marbouty M, Cournac A, Flot JF, Marie-Nelly H, Mozziconacci J, Koszul R | 2014 | Data from: Metagenomic chromosome conformation capture (meta3C) unveils the diversity of chromosome organization in microorganisms | doi.org/10.5061/dryad.gv595 | Available at Dryad Digital Repository under a CC0 Public Domain Dedication. |

## Animations

**Animation 1**. 3D reconstruction of the bacterial meta3C contact matrix.

**Animation 2**. 3D reconstruction of the *E. coli* genome with a plasmid F' carrying a 140 kb segmental duplication.

**Animation 3**. 3D reconstruction of the *E. coli* genome with a plasmid F'.

**Animation 4**. 3D reconstruction of the *B. subtilis* genome.

**Animation 5**. 3D reconstruction of the *V. cholerae* genome.

**Animation 6**. 3D reconstruction of the *V. cholerae* genome.

**Animation 7**. 3D reconstruction of the 11 yeasts meta3C contact matrix.

**Animation 8**. 3D reconstruction of the *Y. lipolytica* genome.

**Animation 9**. 3D reconstruction of the *N. castellii* genome.

**Animation 10**. 3D reconstruction of the *N. castellii* genome after meta3C de novo assembly and GRAAL scaffolding.

**Animation 11**. 3D reconstruction of the *S. bayanus* genome after meta3C de novo assembly and GRAAL scaffolding.

**Animation 12**. 3D reconstruction of a genome closely related to *A. veronii*.

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
