## [Decision Letter]

Thank you for sending your work entitled “meta3C unveils the diversity of chromosome organization in microorganisms” for consideration at *eLife.* Your article has been favorably evaluated by Detlef Weigel (Senior editor) and 3 reviewers, one of whom is a member of our Board of Reviewing Editors.

The Reviewing editor and the other reviewers discussed their comments before we reached this decision, and the Reviewing editor has assembled the following comments to help you prepare a revised submission.

This paper presents the innovative idea of using Hi-C technology to aid the deconvolution of metagenomic samples. Through a series of experimental manipulations the authors investigate the potential of the method for mixtures of bacterial and yeast species. An application to a cultured environmental sample shows that the method can recover structure within real samples. As indicated by the title, the authors also use the results to characterise diversity in certain aspects of chromosome organisation among microbes.

This work goes beyond previous work (Burton et al. G3, Beitel et al PeerJ) on the use of Hi-C in metagenomics, adding a wider series of trial experiments and application to an (albeit cultured) environmental sample. All reviewers feel that the work shows promise, but the reality of using the technique on true environmental samples is still some way off. Nevertheless, this is a valuable addition to this new field and will be of wide interest.

In revising the manuscript, there are a few key areas to address:

1) The hypothesis of integration of the plasmid into the *E. coli* genome needs to be validated using an independent experiment. This is a potentially important demonstration of the technique's ability to identify structure and needs to be validated.

2) We would like to see a deeper and more quantitative evaluation of the similarities/differences between the method presented here and the Beitel/Burton methods.

3) We strongly encourage the authors to either evaluate a truly environmental sample without culturing, or discuss the steps needed to get to this position. This is a major limitation of the method as it stands.

Further detail is provided below:

*Reviewer #1*:

1) I think the hypothesis of F plasmid integration within the *E. coli* genome should be validated experimentally.

2) I would like to see some discussion of whether temporal structuring arising from replication could lead to the contacts observed. Specifically, I wondered whether this might explain the X pattern seen between *V. cholera* chromosomes I and II in Figure 1.

3) It is not clear what is meant by longitudinal organisation.

4) Reference should be to Figure 1, not D.

5) Figure 2 – it's not clear what is meant by invisible reads.

6) I feel that the analysis of the *S. banyans* assembly is rather superficial as there is no truth data to compare it to. I would like to see side by side an analysis of how well the approach can identify and assemble de novo the genome of a species where the genome is known well. This seems to me to be the right test for the method.

7) What is meant by background interactions between communities?

8) The *A. veronii* analysis (rather like most of Figure 1) relies very heavily on mapping back to a reference genome. I feel this is a rather weak ending to the paper; you really want to be able to show that the metaC approach doesn't rely on existing external knowledge to this degree.

9) It's not clear to me why contigs > 2.5kb have to be broken up? This seems bizarre.

*Reviewer #2*:

1) How was taxonomy assigned in the test environmental metagenomic communities? Did all contigs end up being of the same consensus taxonomy? What proportion shared a consensus taxonomy?

2) What fraction of reads go to unmapped/unresolved/not used in the assembly?

3) In albeit simple communities; this method seems to require super deep sequencing to get both HiC reads and enough reads for normal assembly.

4) Are there unique challenges or advantages to performing HiC on enucleated cells?

5) For Figure 1, I would suggest a standardization of colors. 2C, 2D uses blue for *Bacillus subtilis* but 2E uses green for *B. subtilis*. For 2B should consider which bacteria are rods versus spherical and use appropriately.

Reviewer #3:

1) Can the authors discuss why they think Figure 2 is so different from Figure 1? Is it because the genomes of yeast are bigger? Because more species are included? Because yeast genomes are more complex?

2) Figure 3 showing that authors can capture the chromosome conformation in a natural environment would be better served as one of those “space-filling dot plots” used in Figure 1 of the chromosome conformations of the three organisms.

---

## [Author Response]

*1) The hypothesis of integration of the plasmid into the E. coli genome needs to be validated using an independent experiment. This is a potentially important demonstration of the technique's ability to identify structure and needs to be validated*.

We thank the referees for encouraging us to have a second look at this structure. We have performed an in-depth analysis of the behavior of the plasmid F and revised our interpretation. Since the plasmid exhibited frequent contacts with a discrete region of the *E. coli* chromosome, we initially concluded that the plasmid had become integrated. However, upon closer examination of the sequence coverage we realized that the large (170kb) region of the *E. coli* chromosome that was in contact with the plasmid appeared duplicated in that strain. This raised the possibility that the duplicated chromosomal region showing high contact frequency was actually carried by the F plasmid. Analyzing the contacts between the plasmid F specific sequence and the *E. coli* genome we could identify a high contact correlation between the plasmid and the duplication. Such similarity of behavior with such sharp drops at the boundaries clearly indicated that the duplication was carried by the plasmid, which we confirmed using PFGE and Southern hybridization of specific probes.

This observation emphasizes the difficulty to handle DNA duplications when describing genomic structures through those approaches. Even with the power of HiC-based techniques to resolve genome assemblies, such events can introduce misleading signals that needs to be carefully deconvolved. But, importantly, the 3D signal also really helps interpreting those datasets.

In addition, we observed a significant increase in contacts between the F plasmid and the *Ori* and, to a lower extent, the *Ter* regions of the *E. coli* genome, suggesting an average preferential position of this plasmid near these chromosomal regions. This analysis is the first that characterize the average disposition of an episome in a cell through a genome-wide 3C approach. The result is in agreement with previously published data obtained from cell imagining, as we now describe in the manuscript.

We have written a new paragraph describing this analysis and added a new figure to illustrate the results.

“The *E. coli* HB101 strain used in the analysis carries a plasmid F (Figure 1, black arrow). The 3D data provided us with the opportunity to carefully investigate the structure and spatial positioning of this plasmid with respect to the *E. coli* chromosome. We first noticed that a large (∼140kb) region of the *E. coli* genome exhibited a two-fold increase of the read coverage, indicating a segmental duplication. This region was also enriched in contacts with the F plasmid sequence (Figure 1—figure supplement 2), prompting us to hypothesize that one of the two copy was carried by the plasmid. A correlation analysis of the contacts between these two regions of interest (a chromosome region encompassing the duplication, and the plasmid) revealed clearly that, on the one hand, a copy of the duplication is in close contact with the F plasmid (with an IS2 at one the boundaries [coord. 390,063] and an IS3 within), and that on the other hand the plasmid contacts with the *E. coli* genome drop sharply past the duplication boundaries, in agreement with an integration of one copy of the duplication within the plasmid (Figure 1—figure supplement 2). This hypothesis was confirmed experimentally by a Southern blot of a pulsed-field gel (Figure 1—figure supplement 2). Having deciphered the linear structure of the F plasmid within this strain, we investigated its 3D organization with respect to the *E. coli* genome (Figure 1—figure supplement 2). The frequent contacts each copy of the duplication makes with either the chromosome and the F plasmid resulted in an artifactual colocalization of the plasmid with the chromosome, since the contacts within each of the copies cannot be discriminated and are all positioned along the genome (see Figure 1—figure supplement 2, i and Video 3). To solve this bias, the contacts involving the duplicated region were removed and the 3D structure of the resulting was generated (Figure 1—figure supplement 2, ii and Video 2). In the resulting structure, the bin representing the plasmid appeared now well isolated from the chromosome (Figure 1). As a supplementary control, we verified that removing from the genome a region of a size similar to the duplication did not impair the 3D positioning a DNA segment of a size similar to the F plasmid, positioned within this deleted region (blue dot; Figure 1—figure supplement 2, iii). Interestingly, besides this duplication, two regions along the *E. coli* chromosome appeared enriched in contacts with the F plasmid: the replication origin (*Ori*) and, to a lower extent, the termination (*Ter*) regions (Figure 1—figure supplement 2). This that the F plasmid is positioned preferentially in the vicinity of these regions in fast growing cells, as expected given the preferred location of the F plasmid in mid, 1/4 and 3/4 positions in the cell, which are also preferred positions for *Ori*/*Ter* regions during the cell cycle (16; 39). These results illustrate the high specificity of the meta3C approach, which allows the identification of genomic regions belonging to each species, and the power of using DNA physical contacts to decipher complex structures in individual genomes.

*2) We would like to see a deeper and more quantitative evaluation of the similarities/differences between the method presented here and the Beitel/Burton methods*.

We have added a discussion regarding the similarities/differences of our study compare to those from Beitel/Burton. The Burton approach involves prior knowledge of the content of the tested mix as well as of the number of chromosomes of the species analyzed. The ability of our approach to successfully characterize, through a blind analysis, communities of contigs generated directly from the reads is now emphasized in the Discussion. In addition, we identified and described limitations in our analysis that both studies are probably facing but did not mention, such as the complications introduced by chimeric contigs.

Another important difference is the specificity of the experimental protocol developed in our analysis. We believe that we have characterized good conditions to perform HiC in microorganisms as evidence by our results that can be used for 3D reconstruction.

“Taking advantage of chromatin conformation capture data to address genomic questions is a dynamic field: while this paper was under review, two studies were released that also aimed at exploiting the physical contacts between DNA molecules to deconvolve genomes from controlled mixes of microorganisms (3; 8). Although these analyses open interesting perspectives, the meta3C approach presented in this work differs in several aspects. First, both studies used HiC to scafold genomes from contigs obtained either from simulated assemblies (3) or from independent experiments (8). Our approach conveniently uses a single meta3C library and a blind analysis for generating contigs, binning them, scaffolding them and revealing the 3D structure of the corresponding species. Such approach based on a single experiment may require more sequencing depth than an approach combining multiple libraries (HiC + shotgun + mate pair, for instance), but the exact trade-off remains to be determined and depends on the specifics of the experiment, the sequencing technology, and on the amount of starting material available.

Besides, our blind analysis allowed us to delineate communities without prior knowledge of the number of species present in the mix; this contrasts with the approach of [8], which apparently requires such prior knowledge. Similarly, we generate de novo chromosome scaffolds without any assumptions regarding the number of chromosomes present in the mixture, a realistic approach when it comes to exploring environmental samples or complete the assembly of complex genomes.

Furthermore, our experience with the analysis of chromosome organization in both bacteria and yeast species emphasized the importance of the initial cross-linking step for the success of such experiment. The adequacy between the cross-linking step, which depends on the incubation time and the concentration of the fixating agent, and the restriction step and choice of the restriction enzyme, is essential for the recovery of long-distance *cis* contacts that improve de novo genome scaffolding and, importantly, reveal the 3D structure. Optimizing the cross-linking conditions allows new insights into the diversity of 3D chromosome organization of several species.”

*3). We strongly encourage the authors to either evaluate a truly environmental sample without culturing, or discuss the steps needed to get to this position. This is a major limitation of the method as it stands*.

This is an important point that we are planning to address in the near future, in the continuity of the current analysis. We have added a paragraph in our article to discuss how we envision that such experiments should be conducted.

“Last but not least, the meta3C approach remains to be applied on a truly natural (not enriched) sample, such as a gut or wine microbiome. Based on our experience, we envision that, this experiment should include several (two at least) enzymes recognizing sites with various GC percentages, in order for both GC-rich and GC-poor genomes to be appropriately represented in the meta3C library. Doing the experiment with multiple enzymes would also allow taking into account GC content (in addition to average coverage) to improve the binning of contigs into communities. Identifying contigs presenting largely divergent characteristics compared to their neighbors, and redistributing them to their most likely communities would also lead to improve assemblies for each species (see for instance [1]). Overall, quantifying physical contacts between chromosomes provides an objective and convenient principle to segregate the genomes of sympatric species and, from there, to explore the biological diversity of complex ecosystems.”

*Further detail is provided below*:

Reviewer #1:

*1) I think the hypothesis of F plasmid integration within the* E. coli *genome should be validated experimentally.*

See above.

*2) I would like to see some discussion of whether temporal structuring arising from replication could lead to the contacts observed. Specifically, I wondered whether this might explain the X pattern seen between V. cholera chromosomes I and II in*
Figure 1.

This is indeed an interesting discussion. In order to solve this intriguing structure, we have initiated an in-depth analysis of the links between the organization of the chromosomes of *V. cholerae* and the cell cycle of this organism, which goes beyond the scope of the meta3C analysis presented here. However, we have added a new representation of the *V. cholerae* genome 3D reconstruction that clearly illustrates the link between replication and structure. In this figure, the read coverage of the genome, which reflects the average progression of the replication fork and thus the timing of replication, is shown using a color code. Whereas the two *Ter* regions appear of the same color, consistent with the fact that both chromosomes I and II achieve replication synchronously ([43]; Stoke et al., 2011), the *Ori* regions present different colors, with chromosome I initiating replication first whereas chromosome II fires only once the fork has progressed along chromosome I. This analysis illustrates the spatial and temporal control of the replication program of *V. cholerae*. We have added a discussion about this feature in the main text.

“For *V. cholerae*, the *Ter* regions of chromosomes 1 and 2 were in closer proximity than the two *Ori* regions, reflecting the controlled segregation mechanisms of these two unequal-sized chromosomes (51). Interchromosomal contacts did not span the entire length of chromosome 1 but started between the *Ori* macrodomain of chromosome 2 and positions located at about one-third of chromosome 1 arms (black dotted squares, Figure 1). In order to visualize the interplay between replication and genomic organization, we applied a color code to represent the read coverage of the genome (reflecting the average progression of the replication fork and thus the relative timing of replication) atop the genome structure of *V. cholerae* (Video 6). Whereas the two *Ter* regions appeared of the same color, consistent with the fact that both chromosomes I and II achieve replication synchronously (43), the *Ori* regions presented, as expected, different coverages: chromosome I initiating replication first whereas chromosome II starting only to replicate later. This analysis provides a glimpse on the spatial and temporal articulation of the replication program of *V. cholerae*, which would be an interesting example of organization-dependent function.”

*3) It is not clear what is meant by longitudinal organisation*.

We have described what we mean by that more clearly in the text (i.e., that the two replichores extend along parallel axes within the cell).

*4) Reference should be to*
Figure 1*, not D*.

Indeed, thanks.

*5)*
Figure 2
*– it's not clear what is meant by invisible reads*.

We mean by that DNA regions that cannot be disambiguated during the mapping process (i.e. present repeated regions). We have replaced “invisible reads” by the more explicit “remaining contigs”).

*6) I feel that the analysis of the S. banyans assembly is rather superficial as there is no truth data to compare it to. I would like to see side by side an analysis of how well the approach can identify and assemble* de novo *the genome of a species where the genome is known well. This seems to me to be the right test for the method*.

We agree with the referee that this is the best way to assess the performance of our method and have replaced the *S. bayanus* analysis with the *N. castellii* one as a proof of concept. The blind analysis of the reads generated by the meta3C experiment on the mixture of yeast species revealed a community likely to correspond to the genome of *N. castellii*. The GRAAL scaffolding reordered these contigs so that 11 superscaffolds were generated covering 94.5% of the reference genome of *N. castellii*. The results are now described in Figure 2, and Figure 2—figure supplement 3. The same analysis performed on the *S. bayanus* community is now presented in Figure 2—figure supplement 4.

*7) What is meant by background interactions between communities*?

We have detailed what we mean by this term in the sentence:

or because of “background interactions”, which correspond to chimeric religation between restriction fragments belonging to different species resulting from the experimental protocol.

*8) The* A. veronii *analysis (rather like most of*
Figure 1*) relies very heavily on mapping back to a reference genome. I feel this is a rather weak ending to the paper; you really want to be able to show that the metaC approach doesn't rely on existing external knowledge to this degree.*

Ideally, we agree that it would have been better to scaffold the *A. veronii* genome assembly generated de novo from our meta3C library and convert it into a genomic contact map directly, without using the reference genome. However, our coverage of this genome in the meta3C reads was insufficient, which precluded such blind analysis. Nevertheless, mapping the reads on the reference genome allowed us to show that the 3D information needed to perform the scaffolding step is indeed there, albeit in insufficient amount, and that increasing the sequencing depth will alleviate this limitation in future experiments. We now provide a more detailed interpretation of this limitation of our present dataset.

*9) It's not clear to me why contigs > 2.5kb have to be broken up? This seems bizarre*.

We clarified this point by adding in the Methods section the following:

“Before applying the algorithm, contigs longer than 2.5 kb were divided into equal-sized chunks corresponding to several nodes in the graph (since large contigs exhibit more contact than smaller ones, they are therefore more prone to clustering when running the Louvain algorithm).”

Reviewer #2:

*1) How was taxonomy assigned in the test environmental metagenomic communities? Did all contigs end up being of the same consensus taxonomy? What proportion shared a consensus taxonomy*?

The taxonomy annotation was done by analyzing the sequences present in each community of contigs with the software MGRast, which classifies the sequences according to their similarities with known genomes. We have added the MGRast data (Figure 3—figure supplement 1) to illustrate the homogeneity obtained for the largest pools of contigs, and described it in the following text:

“Each community was analyzed using MG-RAST (15), showing relatively homogeneous taxonomic compositions in most communities (Figure 3—figure supplement 1). For each community, more than 80% of the genes identified by MG-RAST within the contigs were attributed to the same taxonomic class. Going down to family level, 8 out of the 11 largest communities were >80% homogeneous, with the remaining 3 presenting more complex patterns that will require further investigation and development.”

*2) What fraction of reads go to unmapped/unresolved/not used in the assembly*?

We have added more precisions regarding these numbers in the Material and Methods section. Since we used a De Bruijn assembly approach, the proportion of unmapped reads is not immediately accessible, and we remapped the reads on the contigs to find out which reads were unaccounted for.

Notably, for the bacterial mixture:

4,040,422 reads were used for assembling the mixture of three bacterial genomes, yielding 2,436 contigs (303>5kb, N50 = 55kb, total length 12.5Mb). More than 95% (∼3,880,000) of the initial reads mapped on the contigs.

For bacterial mix, 10% of the contigs of the smallest size were not attributed to a community with these parameters (representing ∼15% of the total reads).

For the yeast mixture:

47,614 contigs were generated by assembling the ∼60M yeast reads (7,212>5kb, N50 = 6.9kb, total length 138.7Mb). When mapped against the newly generated contigs, 47M reads (∼78%) did align that had been used in the assembly.

For yeast, few contigs were left apart, representing in total ∼2% of the number of reads.

For the river sample:

The assembly generated 130,713 contigs (2,250>5kb, N50 = 1,274kb, total length = 111Mb). After realignment of the initial reads along the generated contig, 42,705,64 PE reads could be mapped against the assembled contigs (∼62%). For the river sediment experiment, 15% of the contigs were left apart during the binning step.

*3) In albeit simple communities; this method seems to require super deep sequencing to get both HiC reads and enough reads for normal assembly*.

These experiments have been made possible thanks to the drop in the cost of sequencing, but the important number of genomes present in natural microbiomes necessitate such large amounts of sequencing to be solved. We have added in the discussion the following comments, in light of the approach chosen by Beitel et al. and Burton et al. (see above).

“Although these analyses open interesting perspectives, the meta3C approach presented in this work differs in several aspects. First, both studies used HiC to scafold genomes from contigs obtained either from simulated assemblies (3) or from independent experiments (8). Our approach conveniently uses a single meta3C library and a blind analysis for generating contigs, binning them, scaffolding them and revealing the 3D structure of the corresponding species. Such approach based on a single experiment may require more sequencing depth than an approach combining multiple libraries (HiC + shotgun + mate pair, for instance), but the exact trade-off remains to be determined and depends on the specifics of the experiment, the sequencing technology, and on the amount of starting material available.”

*4) Are there unique challenges or advantages to performing HiC on enucleated cells*?

Our experience with the analysis of chromosome organization in both bacteria and yeast species push us to emphasize the importance in the initial cross-linking step of the HiC experiment. This step depends on the enzyme used (4 or 6-cutters), the incubation time and the concentration. We have identified good conditions regarding these constraints. This is now clearly stated in the Discussion.

“Furthermore, our experience with the analysis of chromosome organization in both bacteria and yeast species emphasized the importance of the initial cross-linking step for the success of such experiment. The adequacy between the cross-linking step, which depends on the incubation time and the concentration of the fixating agent, and the restriction step and choice of the restriction enzyme, is essential for the recovery of long-distance *cis* contacts that improve de novo genome scaffolding and, importantly, reveal the 3D structure. Optimizing the cross-linking conditions allows new insights into the diversity of 3D chromosome organization of several species.”

*5) For*
Figure 1*, I would suggest a standardization of colors. 2C, 2D uses blue for* Bacillus subtilis *but 2E uses green for B. subtilis. For 2B should consider which bacteria are rods versus spherical and use appropriately.*

We modified the 1B panel accordingly.

Reviewer #3:

*1) Can the authors discuss why they think*
Figure 2
*is so different from*
Figure 1*? Is it because the genomes of yeast are bigger? Because more species are included? Because yeast genomes are more complex*?

We added the following paragraph in the Discussion:

“The average organization of yeast genomes was found to differ radically from those of bacterial nucleoids, most likely because of contrasted replication and division processes and timings. Most of the organization in Saccharomycetes appeared driven by the clustering of the centromeres at the spindle-pole bod; this clustering remains the prominent structural feature throughout most of the cell cycle (i.e., G1 + S phase + G2) during exponential growth, with mitosis representing only a fraction of the cycle. In bacterial, in contrary, the strong overall organizational features are the activation of a unique replication origin per chromosome, the ability to initiate multiple replication forks, and the fast division cycle. In addition, important topological constraints are exerted on bacterial circular chromosomes around the *Ter* regions, and our observation of significant contacts at these positions was probably linked to these phenomena. More imaging and 3C-like analyses will be required to understand the precise choreography of chromosome segregation in these different microorganisms.”

*2)*
Figure 3
*showing that authors can capture the chromosome conformation in a natural environment would be better served as one of those “space-filling dot plots” used in*
Figure 1
*of the chromosome conformations of the three organisms*.

We added a dot-plot of the eleven biggest communities in Figure 3.